# Perspectives from a multi-stakeholder workshop for implementing a risk-stratified breast cancer screening program in Canada
Nareh Safieh-Garabedian [1] ✉, Antonis C. Antoniou[1], Jennifer Brooks [2], June Carroll[3,4], Mathias Cavaillé[5,6,7], Jocelyne Chiquette[6,7,8], MJ DeCoteau[9], Gregory Doyle[10], Andrea Eisen[11,12], Laurence Eloy[13,14], Samantha Feinberg[2,11], Vivianne Freitas[15], Kim Hender[16], Supriya Kulkarni[15], Cynthia Mbuya-Bienge[7,17], Christine Molnar[18], Hermann Nabi[7,17], Caroline Samson[19], Jean M. Seely [20], Amanda J. Sheppard[2,21], Penny Soucy[22], Romeo Tagyen[23], Isabelle Trop[19,24], Annie Turgeon[7], Marc Venturi[25], Nancy Wadden[26], Meghan Walker[2,11], Michael Wolfson[27], Anna Chiarelli[2], Jacques Simard [7,28,29] ✉ & Nora Pashayan [1,29] ✉

Implementing population-wide risk-stratified breast screening (RSBS) requires coordinated planning across diverse stakeholders and adaptation within complex healthcare systems. We convened a multi-stakeholder workshop with 34 policymakers, breast screening program leaders and healthcare professionals across Canada. Participants co-envisioned an RSBS program and focused on defining care pathways, the implementation roadmap, stakeholder engagement, and workforce training. A national, flexible RSBS strategy was proposed with multiple entry points and timings for risk assessment, interoperable information technology systems, phased roll-out, and an embedded learning healthcare system. The resulting actionable plans support RSBS implementation in Canada and offer transferable insights for planning RSBS in varied international settings.

Organized breast screening programs in Canada typically offer biennial screening to women aged between 50 and 74, although many provincial and territorial programs have now lowered the starting age to 40 or 45[1]. Guided by national recommendations, organized breast screening is delivered through provincial/territorial programs, each with its own policies and operational models (e.g., eligibility, entry routes, data systems, and follow-up pathways). All provinces and territories have implemented population-wide breast screening programs, except Nunavut, where breast screening is offered opportunistically, and some jurisdictions have established high-risk pathways (e.g., the High-Risk Ontario Breast Screening Program)[1].

Whilst screening has contributed to a reduction in breast cancer (BC) mortality, it also carries potential harms, including false positives or negatives, psychological distress, overdiagnosis, overtreatment, and increased pressure on healthcare resources[2].

Risk-stratified breast screening (RSBS) could improve the benefit-to-harm balance and enhance the cost-effectiveness of a screening program by tailoring the age to begin and end screening, as well as the frequency and modality to individual risk.

RSBS entails a risk assessment using multifactorial risk assessment tools (e.g., CanRisk)[3–5] to stratify the population into risk categories and tailor risk-stratified screening recommendations. The risk assessment combines information on family history (FH), polygenic risk score (PRS), mammographic density, and questionnaire-based risk information (e.g., reproductive history, age of menarche, menopausal status, body mass index (BMI), alcohol consumption). Among these, the PRS has the strongest influence on how women are stratified by risk[6,7].

Internationally, interest in RSBS is growing. Trials exploring the effectiveness and feasibility of RSBS and comparing this approach to age-based screening include the My Personalized Breast Screening (MyPeBS) large-scale randomized control trial in Europe[8,9] and the Women Informed to Screen Depending On Measures of Risk (WISDOM) study in the United States[10,11].

In Canada, the PERSPECTIVE I&I project (Personalized Risk Assessment for Prevention and Early Detection of Breast Cancer: Integration and Implementation) explored key aspects of implementing personalized breast screening in Canada[12]. This included testing different methods

of inviting participants and communicating risk results. Studies looked into feasibility[13,14], acceptability[15–17], psychological impact[18], organizational readiness for change[12], resource utilization and costs[19], and legal considerations[20,21]. Between 2019 and 2021, 3753 women aged 40–69 in Ontario and Quebec were invited for a comprehensive multifactorial risk assessment using CanRisk to estimate their 10-year BC risk[12,13]. Participants were stratified into average, higher-than-average, or high-risk categories based on age-dependent risk thresholds using relative risk (RR), which is calculated as the ratio of predicted 10-year absolute BC risk to age-specific population risk[22]. Risk results and tailored screening recommendations were provided to participants, demonstrating the feasibility of personalized risk assessment within the current Canadian screening infrastructure[13]. Most participants found learning their risk to be beneficial, and it reduced negative psychological and emotional effects[18]. Delivering RSBS is widely deemed as acceptable by both healthcare professionals[17] and the general public[15] in Canada. In a population-wide survey in Ontario, Quebec, British Columbia, and Alberta, most participants were comfortable sharing genetic and personal information for RSBS[15].

To shift from an age-based screening approach to a risk-based approach, the program should do more good than harm[23], be cost-effective for the healthcare system[22,24], acceptable to both users and healthcare providers[15,16,22], feasible[19], have relevant human and financial resources in place[25], and not worsen health inequalities[25,26]. Further, relevant healthcare organizations need to be ready for change (i.e., willing and able)[27], with all stakeholders engaged throughout the implementation process[28].

RSBS implementation would occur within a complex adaptive system (CAS), a multi-faceted and dynamic network of interconnected components that interact, learn, and adapt over time[29,30]. A systems thinking approach is therefore necessary for implementing RSBS[25], as it views the system as a whole, recognizing interdependencies and continuous change. This approach encourages participatory decision-making and stakeholder engagement while supporting change through iterative learning and adaptation to manage the unpredictable, nonlinear nature of real-world implementation tailored to local contexts[31]. For this, we brought together stakeholders from across different levels of the healthcare system in Canada to plan for implementation. The purpose of our workshop was to explore how to prepare for and implement an RSBS approach within Canadian breast screening programs. In this Perspective, we summarize the outputs of our workshop that covered defining care pathways for a flexible RSBS strategy, developing a phased implementation roadmap, engaging stakeholders, and planning education and training for the workforce. We propose key recommendations for implementing RSBS in Canada that are transferable to other healthcare systems, considering implementation.

## Workshop Design

Our 2-day workshop in Quebec City took place on October 30–31st 2024, involving 42 attendees, including 34 invited participants and 8 facilitators, from across Canadian provinces and territories (Supplementary Table 1). Participants were stakeholders involved in the delivery of breast screening programs or in policy decision-making roles. We use "stakeholders" as a practical shorthand for interested and affected organizations, groups, and individuals within the health system. Many participants held multiple roles, including individuals working in provincial governments ($n = 14$), managers or directors of jurisdictional breast screening programs ($n = 17$), and healthcare practitioners ($n = 8$), including family physicians and radiologists. One participant attended as a patient/public advocate, and two participants also represented patient/public organizations, including the Quebec Breast Cancer Foundation and Rethink Breast Cancer.

Furthermore, almost half of the participants were representatives of the Canadian Breast Cancer Screening Network (CBCSN), a community of practice made up of partners responsible for the delivery of BC screening across Canada and tasked with improving the quality, equity, and delivery of population-based breast screening services across Canada (Supplementary Table 2). The CBCSN is hosted by the Canadian Partnership Against Cancer

(CPAC), which is a federally funded organization collaborating with partners across the country to implement the Canadian Strategy for Cancer Control[32].

In adopting a systems thinking lens, we incorporated design thinking principles to encourage creative, stakeholder-driven collaboration. We drew from the Theory of Change framework[33] when planning activities to define vision statements and develop objectives and actionable plans to prepare for implementation.

The workshop began with the presentation of key findings from the PERSPECTIVE I&I study to establish a baseline understanding of RSBS planning in Canada. This used the CanRisk tool, but this is just an example of a comprehensive multifactorial risk assessment tool, and other validated tools may be considered. Following this, there were four activities with six breakout discussions, each followed by working groups reconvening to share with the main group (see Supplementary Table 1).

The workshop was approved by the Research Ethics Board of the CHU de Québec-Université Laval (#2021-5136), and all participants provided informed consent to participate.

## Mapping the pathway for a risk-stratified breast screening (RSBS) program

Implementing RSBS requires introducing a population-wide systematic multifactorial risk assessment. Most breast screening programs across Canada already collect some information on BC risk factors at the time of screening[32]. For RSBS, the current data collection would need to be expanded to incorporate the new polygenic risk score (PRS) component and streamline the processes. Table 1 broadly compares the differences between the steps involved in current breast screening programs and an RSBS program.

Drawing on workshop discussions, two possible pathways were considered for offering RSBS. In the first option (Fig. 1a), women aged 40 would complete a comprehensive risk assessment during their first screening mammogram, and the resulting risk category would inform the future screening interval and imaging modalities. Alternatively (Fig. 1b), risk assessment would be offered *before* the age of 40. Here, PRS and other risk factor information would estimate individual BC risk and assign the individual to a risk category to determine what age to have the first mammogram (i.e., earlier or later than age 40). Then, after the first mammogram is done, breast density would be added to recalculate risk and determine the screening interval and imaging modalities. As some risk factors change over time (e.g., BMI and FH), risk assessment needs to be repeated periodically. In either pathway, risk assessment could be done in primary care or radiology facilities.

Results could be delivered in writing (e.g., via letter or online portal) or also combined with a consultation (e.g., in person or with a virtual counselor[34]). High-risk individuals may receive an initial invitation for counseling or a letter with the option to request follow-up support. In the PERSPECTIVE I&I study, results were sent to both individuals and primary care practitioners (PCP) in Quebec, whereas in Ontario, participants chose whether to discuss results with their PCP. Regardless of risk level, all participants could request a phone consultation with a genetic counselor[14]. Furthermore, healthcare professionals in the study reported the importance of providing clear, easy-to-understand results with additional links to online resources if needed[17]. Jurisdictions should decide how results are communicated and ensure accessibility to multilingual, literacy-appropriate resources to support diverse populations.

Jurisdictions need to determine logistical considerations around governance bodies, information technology (IT) systems, and data linkages to support RSBS processes. These could depend on existing infrastructure, workforce capacity, and workload for follow-up care. Participants generally found it most feasible for breast screening programs to govern the administrative processes, including risk score data entry and ensuring risk letters are generated and shared with individuals and their PCP, building on existing invitation and communication mechanisms (Table 1). An integrated IT infrastructure with interoperability will be essential to link data

**Table 1 | Current breast screening program vs RSBS, what would need to change, and whether these changes are perceived as feasible or more challenging**

| Current breast screening programs | Risk-stratified breast screening programs | What would need to change | Perceived: feasible vs more challenging[b] |
|---|---|---|---|
| **(1a) Target population identification and invitation** | | | |
| **Identification of the target population** | | | |
| Average-risk women aged 50–74: typically, a biennial mammogram; many jurisdictions lowered the starting age to 40 | Population-wide risk assessment: entry at age 40, time of first mammogram screening (Fig. 1a), or aged between 30 and 39 (Fig. 1b) | Population-wide systematic risk assessment | *Feasible*: population registries allow identification of eligible participants |
| **Invitation** | | | |
| Entry for mammogram varies by P/T: typically includes referral from primary care; referral from radiology; program invitation; self-referral[41] | Invitation for risk assessment through the same current breast screening/ surveillance programs | Update invitation content to include risk assessment information, saliva sample kit for PRS testing, and questionnaire | *Feasible*: same entry points; existing invitation systems/infrastructure, including invitation mechanisms (letters, portals, digital communications, etc.). *More challenging*: logistics; communications; time |
| **(1b) Risk assessment through follow-up screening** | | | |
| **Risk assessment** | | | |
| Some P/Ts have programs or pathways for high-risk individuals Many already collect information on some risk factors, but *not* for a formal risk assessment[1] | Risk assessment entails: PRS; questionnaire-based risk factors (e.g., family history, BMI, alcohol consumption, age at first childbirth, parity, history of cancer); mammographic density | Offer risk assessment to everyone and stratify into risk categories Develop interoperable IT infrastructures to collate data and transfer to the risk assessment tool (e.g., CanRisk) Need a genetic test performed by accredited labs for PRS | *Feasible:* PRS testing with saliva sample collection - experience with home kits (e.g., FIT). *More challenging*: setting up enough accredited laboratories for PRS; integrating CanRisk into EMR; entering risk information from multiple sources into CanRisk seamlessly; healthcare professional capacity; having policies and regulations to prevent exacerbation of health inequalities |
| **Risk assessment result** | | | |
| Individuals with a strong family history of cancer may be referred for genetic counseling and further testing for pathogenic variants, e.g., *BRCA1/ BRCA2*[41] | Risk assessment result for risk stratification Jurisdiction to define the number of risk categories | Assign individuals to risk categories and link to tailored screening recommendations | *Feasible:* evidence supports that RSBS would be acceptable to the majority of women in Canada[15] |
| **Risk communications** | | | |
| By a genetic counselor for high-risk individuals | Provide risk estimate and screening plan to the individual or the individual and/or primary care practitioner (PCP) Results given as absolute risk value and/or risk category | Train healthcare professionals to interpret and communicate risk results and screening recommendations | *Feasible*: for risk results given in writing and counseling limited to the high-risk group; disseminate accessible informational resources about the risk category and screening recommendations. *More challenging*: managing healthcare professionals' time and capacity |
| **Screening test** | | | |
| Biennial mammograms for average-risk women. Some programs have pathways for high-risk women that offer ages 30–74 annual mammogram and breast MRI (if advised)[41] | Tailor screening to the risk category | Design care pathways and resource plans to match screening recommendation demands | *Feasible*: already have much infrastructure —most jurisdictions offer biennial mammograms as well as MRIs in some cases. *More challenging*: need evaluation of potential mismatches between demand and resources (i.e., MRI machines and radiologists) |
| **Screening result** | | | |
| Results sent to individual and/or PCP by mail or via online portal (varies by P/T)[1] | Same | Same | *Feasible*: continuation of existing processes |
| **Follow-up diagnostics** | | | |
| Abnormal diagnostic test: women with BC followed by surveillance Normal result: invite for next mammogram (ages 40–74)[41] | Same | Same | *Feasible*: continuation of existing processes |
| **Follow-up screening** | | | |
| Routine re-invitation for next mammogram (ages 40–74)[41] | Individual follows personalized screening recommendations Reassess risk level as some risk factors change over time | Reinvite the individual for risk assessment | *Feasible*: mirror the initial workflow. *More challenging*: logistics for repeat risk reassessment |

**Table 1 (continued) | Current breast screening program vs RSBS, what would need to change, and whether these changes are perceived as feasible or more challenging**

| Current breast screening programs | Risk-stratified breast screening programs | What would need to change | Perceived: feasible vs more challenging[b] |
|---|---|---|---|
| **(1c) Performance monitoring and information systems** | | | |
| **Monitoring of program performance** | | | |
| Each P/T monitors and evaluates its program. CPAC collates and analyses data (environmental scans) across P/Ts[41] | Same | Same | *Feasible*: continuation of existing processes |
| **IT systems and data linkages** | | | |
| P/T breast program retains data ownership[a] and submit evaluation indicator data to CPAC Standardized reporting, data linkages, and regional registries vary by P/T[69] | IT system collates risk assessment data from radiology, laboratories, and questionnaire responses into one system linked by unique record IDs. System transfers data into the risk assessment tool (e.g., CanRisk) and returns risk category results, and automates a letter for the individual and PCP (Fig. 2) | Interoperability between IT systems and advanced infrastructure | *More challenging*: sharing data between health facilities, depending on existing IT infrastructure, due to decentralized healthcare systems |

*RSBS* risk-stratified breast screening, *PRS* polygenic risk score, *BC* breast cancer, *EMR* electronic medical record, *P/T* province/territory, *MRI* magnetic resonance imaging, *IT* information technology, *FIT* fecal immunochemical test, *CPAC* canadian partnership against cancer.
[a]Includes participation rate, retention rate, abnormal call rate, diagnostic tests, and cancer stage at diagnosis, etc.
[b]Feasible and more challenging indicate workshop participants' perceptions of the relative ease/complexity of implementing each change.

from multiple sources into one system that feeds into a risk assessment tool (Fig. 2).

Workshop participants emphasized that individuals should have the option to decline risk assessment. In such cases, an alternative pathway should be available, such as sustained access to standard age-based screening (with appropriate information and the option to reconsider risk assessment in the future). Participants also noted that existing high-risk identification and management pathways (e.g., for individuals with known pathogenic variants such as *BRCA1/BRCA2*) are distinct from the population-wide RSBS pathways described here. As high-risk identification and services differ across provinces and territories in Canada (formal programs vs pathways), Fig. 1 illustrates key pathways for RSBS and is not intended to be exhaustive. High-risk pathways were not mapped in detail as this was outside the scope of the workshop discussions, although participants considered this an important topic area for future work.

## Defining care pathways
Implementation of RSBS requires clearly defined care pathways. These are evidence-based, systematic processes designed to outline best practices for an individual's healthcare journey. They serve as a structured guide for both healthcare professionals and the public, ensuring consistency in the care individuals receive[35].

In creating RSBS care pathways, the aspiration of the workshop participants was to "*transform more women with symptomatic breast cancer to screen-detected breast cancer and integrate all risk categories into one screening program to provide equitable access to all Canadians. The aim is to drive innovation of personalized screening and empower individuals to make informed decisions, ultimately transforming healthcare for all Canadians.*"

Pathways should specify how individuals are invited for risk assessment, the required steps, where assessments occur, and which healthcare professionals are responsible. They should ensure interoperability of IT systems, defining how risk assessment data are collected, integrated, and securely shared across platforms, and outline governance structures linking risk results to tailored screening recommendations.

There was consensus among participants to use a comprehensive multifactorial risk assessment. Individuals could complete the risk-factor questionnaire at home or on-site, either online (via web link or digital device), or in paper-based format with trained practitioner assistance if needed. For PRS analysis, saliva samples could similarly be collected either at home or on-site and sent to an accredited laboratory for genotyping. Each risk assessment component (PRS, questionnaire responses, and breast density) must be collated from respective sources and linked to a unique

record ID (Fig. 2). Individuals could either be mailed a paper questionnaire or emailed an online link along with a saliva kit to complete at home before attending a separate mammogram appointment (Fig. 2a). Alternatively, all components could be collected during a one-stop-visit, for example, at a radiology facility (Fig. 2b). In the latter, the mammogram, saliva kit and questionnaire would be completed during the same appointment.

The PERSPECTIVE I&I project showed that RSBS pathways were feasible, building on existing infrastructure. In Quebec, questionnaire responses were automatically formatted into CanRisk-compatible structures and entered into the system. Radiology facilities were contacted to fax individual breast density to the research team, and PRS results were sent from an accredited lab directly to the research team, which was responsible for entering all data into the CanRisk-compatible structure and coordinating any necessary follow-up or verification with individuals. A cloud-based prototype system was developed to transfer risk assessment data from REDCap to PULSAR's OPAL database, which securely transferred the data as batches into CanRisk via Application Programming Interface (API)[4]. Results were then returned under a unique record ID. Automated risk letters were generated, reviewed by the research team, and shared with the individual and their PCP.

## Perceived opportunities/challenges and risks/mitigations in defining care pathways to implement RSBS
The PERSPECTIVE I&I study has already laid the foundation for defining care pathways for RSBS. Workshop participants raised this as an opportunity and highlighted Canada's well-established quality assurance frameworks and feedback mechanisms to support pathway development and refinement over time. Other potential opportunities discussed include leveraging electronic medical records (EMRs), interoperable digital health platforms, and emerging AI tools to reduce administrative burdens on healthcare providers[36].

Challenges were discussed around designating responsibilities and managing workloads for healthcare professionals who already have limited capacity. To mitigate this, participants recommended designing pathways that distribute responsibilities across different roles to prevent over-burdening any single group. Other proposed solutions included expanding the clinical workforce by involving allied healthcare professionals, such as nurse practitioners, pharmacists, or trained volunteers for support. Broad and flexible pathways would also allow jurisdictions to tailor roles, responsibilities, and workflows to their existing capacity.

Participants expressed concerns about the potential impact of risk categorization on individuals' lives (e.g., insurance premiums). In a survey,

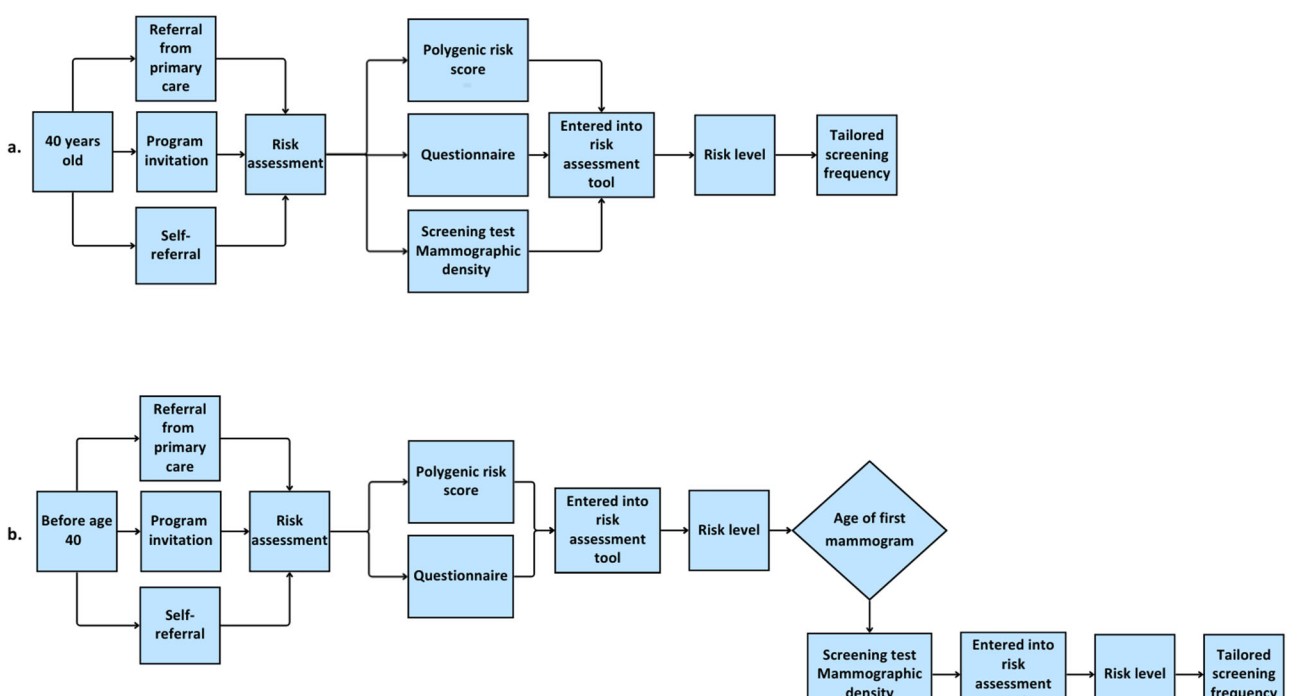

**Fig. 1 | Suggested pathways for RSBS.** Suggested pathways for risk-stratified breast cancer screening. **a** Age 40, the individual has a risk assessment at the same time as the first mammogram to determine the screening interval. **b** Before age 40, an individual has a risk assessment using PRS and questionnaire-based risk factors to determine at what age to have the first mammogram.

approximately one-third of Canadian women reported worries about employers' or insurers' access to genetic risk information, which would need to be addressed in implementation planning[37]. In defining care pathways, participants emphasized the need to protect everyone from any potential health stigmatization and discrimination based on their risk category, recommending strong privacy safeguards and policy regulations to maintain public trust and ensure equitable participation in risk assessment and care pathways. A recent legal study revealed further clarification is needed on the extent to which genetic risk information from prediction models would be protected under the Genetic Non-Discrimination Act (GNDA)[38]. Additionally, public education campaigns were recommended to improve awareness of individual rights and protections[21].

The initial adoption of processes and structures may inevitably introduce unforeseen challenges or errors, such as risk misclassification, miscommunications, or IT system failures. Jurisdictions should establish clear accountability protocols, specifying responsibility and financial liability in such circumstances. Other mitigations include maintaining transparency in defining all processes, ensuring up-to-date documentation, conducting regular reviews, and establishing legal agreements that define financial responsibility and protocols.

**Action plan for defining care pathways**
Participants preferred to have a pan-Canadian strategy developed in collaboration with federal, provincial, and territorial stakeholders with standardized RSBS guidelines to ensure consistency in risk category definitions and screening recommendations. However, the logistics (i.e., how invitations are sent and how risk results are communicated) could be adapted at the jurisdictional level to align with healthcare system capacities. Pathways should therefore be broadly defined yet flexible. Participants suggested using the PERSPECTIVE I&I project protocol[12] as a draft starting point for planning.

RSBS pathways would require interoperable IT systems to support data linkages across multiple sources, seamless integration with EMR, and alignment with clinical workflows. Fit-for-purpose infrastructure would be essential, including the establishment of a genetic test by accredited

laboratories for PRS genotyping. Careful planning around legal, privacy, and ethical considerations is critical, with policies and regulations needed to ensure confidential and secure data linkage protocols to protect personal information[26].

Findings from the PERSPECTIVE I&I project revealed that some population groups, such as individuals not born in Canada and some minority groups, required additional resources and support to complete risk assessments[13]. Therefore, pathway designs should incorporate culturally sensitive targeted strategies to ensure equitable access to *all* population groups. Governance frameworks could further support compliance with privacy laws and regulations and help prevent worsening genetic discrimination.

**Implementation roadmap**
An implementation roadmap offers a high-level plan to guide processes from early planning through to implementation and sustained use[39]. It outlines the overarching processes, structures, and resources needed and how these would be obtained. The roadmap should be designed collaboratively by diverse stakeholders to foster shared ownership, build trust, and align priorities in developing strategic objectives[40]. For RSBS, this would include defining care pathways, assessing needs, developing infrastructure such as a genetic test to determine the PRS performed by accredited genetic labs, training healthcare professionals, and developing timelines and milestones. A key strategy would be engaging local champions to raise awareness, build momentum, and support provincial and territorial screening programs.

In planning the roadmap for implementing RSBS in Canada, the aspiration was *"to leverage existing knowledge & resources to provide patient-centered risk-based breast screening that promotes early detection and improves outcomes."*

**Perceived opportunities and challenges of implementing RSBS**
Almost all Canadian provinces and territories have organized breast screening programs with well-established infrastructure, systems, and workflows in place. Some, such as British Columbia, offer high-risk

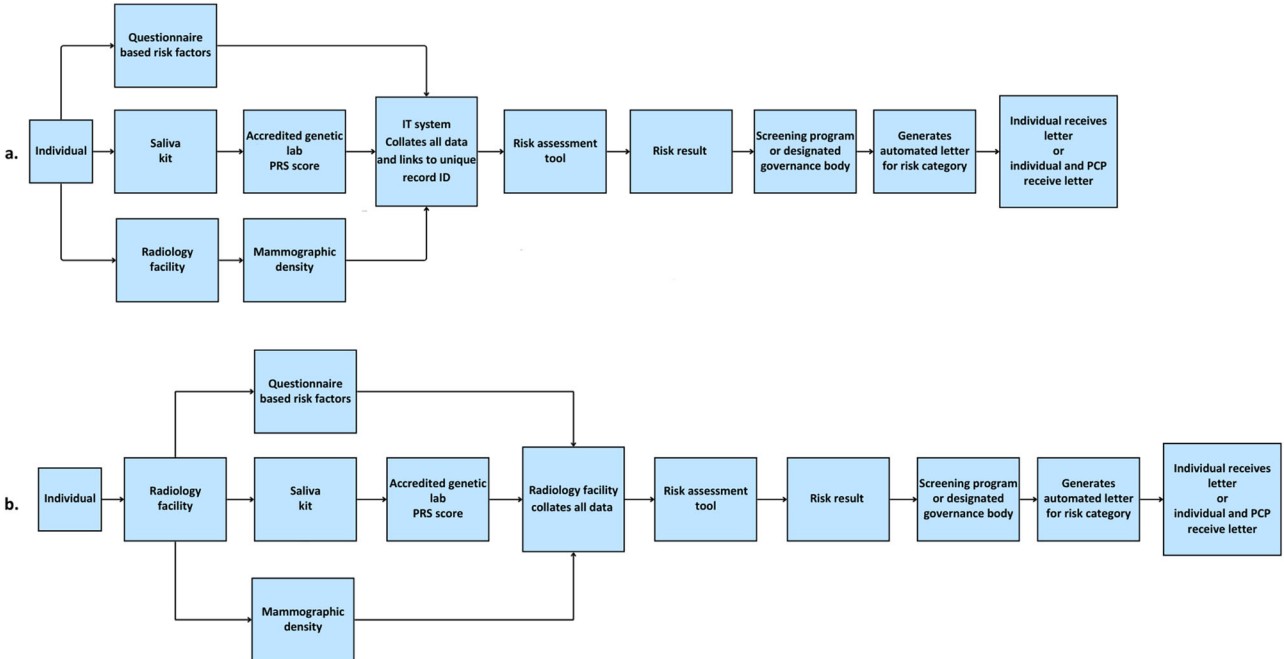

**Fig. 2 | Suggested data linkage pathways for risk assessment either in primary care or a radiology facility. a** *Risk assessment data completed at different time points.* **b** *Risk assessment data collected in one singular visit.*

screening pathways, while Ontario has a high-risk program[41]. The CanRisk tool is user-friendly and already used by healthcare professionals in genetics clinics to identify candidates for high-risk screening[42]. This existing infrastructure can support RSBS integration, reducing the need to build completely new systems from scratch.

The introduction of Canada's lung cancer screening program, which uses risk-based screening, provides a valuable implementation model. Provinces and territories are progressing through various stages of lung screening (i.e., from planning to piloting, partial rollout, and full implementation) following a phased stepwise approach where each stage builds on the last. The CPAC plays a key role in supporting implementation throughout each stage, offering resources, organizing activities, establishing working groups/advisory committees, and supporting pilot studies[43]. For example, CPAC made available resources and tools such as an organizational readiness assessment with partner jurisdictions to support implementation[43]. Furthermore, the Pan-Canadian Lung Screening Network (PCLSN), like the CBCSN, facilitates collaboration and shared learning. Thus, the collective experience of lung screening implementation across Canada offers key insights for RSBS implementation and aligns with the Canadian Strategy for Cancer Control.

To establish an RSBS program from initial planning to full implementation, adequate funding will be critical to allocate resources and ensure IT systems support RSBS processes. Participants identified potential challenges in securing government funds due to competing priorities across other cancer agencies and screening programs. Strengthening the evidence-base for RSBS and developing a compelling business case were considered key facilitators to securing long-term commitment to funding. Furthermore, aligning RSBS with broader national health priorities (e.g., such as embedding it within the Canadian Strategy for Cancer Controls) could further improve funding prospects.

Another challenge would be preparing healthcare professionals with the knowledge, training, time, and resources needed to deliver RSBS while overcoming resistance to change. As mitigations, engaging healthcare professionals and adopting change management strategies were recommended to build buy-in early on and sustain motivation throughout training [see section on Education and Training].

## Action plan for an implementation roadmap

The first step in building the RSBS implementation roadmap would be to assemble a national, multidisciplinary working group to set strategy and oversee implementation processes. The group should include diverse stakeholders such as screening program leads, federal/provincial government representatives, healthcare professionals, patient advocacy groups, First Nations, Inuit, and Métis community and/or organizational representatives, and researchers. During the planning phase, the group could work with provinces and territories to design context-specific business cases, including a needs assessment/situational analysis to identify gaps (Table 2). These would address alignment of priorities, eligibility and participation, evidence strength and quality, workforce training, system-level requirements (financial, infrastructure, digital capacity, and human resources), program monitoring, and health promotion/education. The roadmap would outline action plans to secure these requirements, including workforce training, IT systems, and infrastructure fit-to-purpose, and promoting an environment that is conducive to change.

Participants recommended a phased approach beginning with a pilot implementation to build gradually toward full rollout, allowing potential challenges or modifications to be addressed directly as they arise (Table 2). For example, programs already collecting some risk data, such as FH and mammographic density, could begin by adding the PRS component to existing processes and monitor outcomes before adding additional components incrementally as needed. It could be that the PERSPECTIVE I&I project has already laid the groundwork as a pilot for Ontario and Quebec, other jurisdictions may consider their own pilots to explore feasibility, acceptability, and logistical considerations within local contexts before wider implementation.

The roadmap could adopt a Learning Healthcare System (LHS) approach, recognizing that in a dynamic healthcare system, real-time monitoring and evaluation can inform continuous adaptation throughout implementation[44,45]. Providing stakeholders with timely data would enable context-specific responses to challenges and would support iterative learning to improve the program. For example, monitoring access to risk assessments during a pilot could reveal whether targeted strategies effectively reach marginalized or remote populations, and if not, guide more effective strategies. This would help ensure that care is

**Table 2 | Action plan for developing the RSBS implementation roadmap**

| Stages[43] and actions | Actionable steps |
|---|---|
| *Planning for implementation:*<br>• Assemble national multi-disciplinary implementation group | • Led/co-led by a national body (e.g., CPAC as suggested by participants, given their strong cross-jurisdictional partnerships)<br>• Pan-Canadian governance framework to support accountability, policy alignment, and collaboration |
| • Co-develop a business case with provinces and territories and assess readiness<br>• Situational analysis to assess needs and identify gaps, e.g., infrastructure and workforce capacity<br>Develop an implementation roadmap | • Develop care pathways for RSBS, where logistics around data linkages would be defined<br>• Outline objectives, action plans, roles and responsibilities, timelines, and milestones<br>• Include resource allocation for infrastructure, workforce training, and public education [See section on Defining Care Pathways] |
| • Share with screening program leads | • Circulate briefing pack to provincial/territorial leads (e.g., draft care pathways, governance/IT approach, roles, timelines)<br>• Plan structured consultations (short survey/interviews plus a webinar/workshop) to gather feedback, align priorities, and determine jurisdiction-specific needs<br>• Document and inform: summarize feedback, record decisions, update the roadmap, and keep all stakeholders informed [See section on Stakeholder Engagement] |
| • Develop communication strategies to inform healthcare professionals and the public about RSBS<br>Address potential concerns, e.g., genetic discrimination and data privacy | • Map current RSBS knowledge by healthcare professional role; develop role-specific materials (e.g., scripts, FAQs, consent text); and offer training (e.g., briefings/CME/webinars, quick-reference tools) [See section on Education and Training] |
| *Pilot:*<br>• Design and launch pilot | • Develop a monitoring framework to track and evaluate outcomes<br>• Measure key outcomes (e.g., participation rates and provider/user satisfaction), identify gaps, assess feasibility, and refine implementation strategies |
| *Phased implementation:*<br>• Scale up based on pilot findings<br>• Continuous monitoring and feedback | • Gradual scale-up incorporating context-specific adjustments informed by real-time data<br>• Monitor outcomes, gather stakeholder feedback to improve delivery iteratively |
| *Full implementation:*<br>• Roll-out full program jurisdiction-wide<br>• Continuous quality improvement | • Continuous quality improvement mechanisms to refine systems, structures, and processes accordingly<br>• Establish quality assurance criteria (i.e., define eligibility criteria, patient navigation, participant recall, diagnostic follow-up, etc.) |

*RSBS* risk-stratified breast screening, *CPAC* Canadian Partnership Against Cancer.

delivered equitably, resources are used efficiently, and workflows are continuously optimized.

Scaling RSBS from pilots to broader implementation requires robust quality assurance frameworks and comprehensive, standardized data collection[46]. These systems enable continuous monitoring, feedback, and optimization, ensuring implementation fidelity, equity, and real-world effectiveness comparable to research outcomes[47]. Embedding these mechanisms from the outset supports identification of challenges and data-driven improvements.

## Stakeholder engagement

Successful implementation requires active engagement of all stakeholders in the design, delivery, and evaluation of implementation strategies[28]. Stakeholder engagement is an ongoing, interactive process involving individuals, organizations, and communities affected by or responsible for a decision or policy. It draws on their knowledge, experience, and values to build trust, shared understanding, and support transparent, effective decision-making[48,49]. Engagement approaches can range from unidirectional activities, such as presentations, to more collaborative approaches, including dialog and co-design[50]. During the workshop, participants outlined a list of stakeholders that would be relevant to involve during the engagement processes (see Supplementary Table 3).

In engaging stakeholders to implement RSBS, the aspiration was "*to foster collaborative and inclusive breast cancer screening programs in Canada by implementing universal risk assessment and enabling all stakeholders, patients, healthcare providers, and government agencies to implement personalized screening strategies.*"

### Perceived opportunities/challenges and risks/mitigations in engaging stakeholders for implementing RSBS

Organizations such as the CPAC, provincial and territorial screening programs, the Canadian Association of Radiologists (CAR), and the Canadian Society of Breast Imaging are exemplary bodies that have already established stakeholder collaboration networks in Canada. A pan-Canadian organization like CPAC has the infrastructure, resources, and experience to coordinate national initiatives and could oversee and convene working groups. The group could be organized through existing networks such as the CBCSN, a community of practices that includes representatives from BC screening programs across the country and national health organizations, or by establishing a Canadian Translational Advisory Committee (CTAC).

Challenges were raised around aligning diverse stakeholder views due to competing priorities across jurisdictions and multiple levels of leadership. To mitigate this, an early stakeholder engagement strategy could foster trust and shared ownership of coordinated action through collaborative decision-making activities such as designing objectives that reflect diverse perspectives.

When planning for pilots, a potential risk is that some provincial and territorial breast screening programs may not participate. To mitigate this, participants suggested re-engaging stakeholders, especially at the program level, to gather feedback and adapt pilot plans to better align with regional priorities and needs.

### Action plan for stakeholder engagement

The Implementation-Stakeholder Engagement Model (I-STEM) is one useful practical framework for planning stakeholder engagement to support implementation[28]. For RSBS, once the working group is assembled, the first step would be stakeholder mapping to identify all relevant stakeholders who are affected by or can influence implementation (Table 3). Participants emphasized the importance of an inclusive strategy, ensuring representation of marginalized or less visible groups in decision-making processes. Remaining open to feedback and flexible in engaging new stakeholders throughout the process can further enhance representativeness[51]. The group should refine mapping criteria and update the stakeholder map as roles, influence, and priorities evolve, and throughout the different parts of the implementation roadmap.

**Table 3 | Actionable steps in engaging stakeholders for RSBS, drawing from the I-STEM model**

| Step | Action | Steps |
|---|---|---|
| Step 1 | Stakeholder mapping | Implementation group: identify all stakeholder groups that would be affected by/influence the implementation of RSBS for an inclusive approach |
| Step 2 | Develop engagement objectives | Develop engagement objectives that define the purpose of engaging each group<br>Examples: designing pathways and logistics, building support, gathering input, addressing concerns |
| Step 3 | Stakeholder analysis | Determine stakeholder analysis approach<br>Example: Power-interest approach classifies stakeholders to strategically tailor engagement efforts towards different groups[51]<br>• Players (high power, high interest)<br>• Context setters (high power, low interest)<br>• Subjects (low power, high interest)<br>• Crowd (low power, low interest) |
| Step 4 | Define engagement approach | The type of work to be done with stakeholders to achieve engagement objectives<br>Four core approaches:<br>• Disseminating information (e.g., health promotion/education)<br>• Assessing perspectives<br>• Consulting stakeholders<br>• Collaborative design<br>   Would be determined by engagement objectives, resources availability, political priorities, and stage of implementation roadmap |
| Step 5 | Define qualities and logistics of engagement approach | Determine logistics for carrying out activities to achieve engagement approach by qualities such as stakeholder 'preparedness', activity 'structure', extent of stakeholder engagement in implementation work, 'regularity' of engagement, and 'accountability'<br>Examples of activities:<br>• Newsletters and reports<br>• Surveys and interviews<br>• Feedback sessions<br>• Workshops or planning sessions |
| Step 6 | Review engagement outcomes | Review how engagement outcomes link back to engagement objectives<br>Soft outcomes (stakeholder perceptions of engagement) vs. hard outcomes (tangible results)<br>Regularly refer to the implementation roadmap to maintain engagement objectives and activities across all stages of implementation |

*RSBS* risk-stratified breast screening, *I-STEM* Implementation-Stakeholder Engagement Model.

The implementation group would then develop engagement objectives in terms of purpose for each step of the roadmap. Stakeholder groups should be analyzed using pre-defined criteria such as power, interest, expertise, impact, capacity, trust, and orientation towards RSBS to understand each group's concerns, influence, potential impact, and required level of engagement at different stages. Based on this, the group would define engagement approaches, outlining the type of work to be done with stakeholders to achieve the objectives, as well as their quality in terms of how the engagement would be planned and carried out. Once engagement activities are scheduled, relevant stakeholders should be involved according to those timelines.

Given the diversity of stakeholders and objectives, hybrid engagement approaches may be necessary at different stages of implementation. For example, information resources could be distributed to medical communities, while consultation and co-design sessions could engage national decision-makers, public health officials, and provincial screening program leads to develop RSBS guidelines.

Ongoing, iterative evaluation of the engagement objectives and activities would be essential as stakeholder perspectives may shift, new evidence and information may emerge, and leadership or staffing could change. Ultimately, by incorporating diverse perspectives, addressing concerns, and keeping all groups informed, implementation efforts are more likely to succeed[28].

## Education and training for healthcare professionals

If healthcare professionals involved in breast screening (e.g., primary care, radiologists, and nurse practitioners) are to lead risk assessments and communicate results, studies have shown that their current knowledge of RSBS and PRS testing is insufficient[52–54]. As such, specific training would be essential to equip them with the skills to use risk assessment tools, interpret risk categories, and communicate results effectively[54], helping build trust and support shared decision-making[55]. Training would cover risk interpretation and effective communication strategies, and would also include

the effects of common and rare genetic variants, pathogenic variants with uncertain risk estimates, Variants of Uncertain Significance[25], and protocols for discussing such information[54]. However, concerns remain about increased workload, limited time, and strained resources among all healthcare professionals[16,17,56,57]. A well-designed training and education strategy would be essential to prepare healthcare professionals for RSBS while accommodating capacity and time constraints.

In training and educating the healthcare professionals to implement RSBS, the aspiration was "*to clearly define the roles, responsibilities, and tasks for healthcare professionals, specifying what each role is accountable for. Develop strategies for workforce education and training, outlining what this training will entail. Guiding principle to acknowledge varying levels of current knowledge and aim to establish a foundational level of understanding for all.*"

### Perceived opportunities/challenges and risks/mitigations in training healthcare professionals to offer RSBS

Participants identified opportunities to enhance training by integrating RSBS into existing frameworks, such as Continuing Medical Education (CME), using webinars or online modules. For example, this could build on existing models such as Massive Open Online Courses (MOOCs) for genetic variant interpretation and risk communication skills, and adapt "just-in-time" learning tools to each national context[58].

Nationwide multidisciplinary forums could also support knowledge sharing and resource development for RSBS, similar to the UK's National Cancer Genetics and CanGene-CanVar Program, which has facilitated discussions on complex genetic profiles, expanded clinical knowledge, and enhanced patient care in cancer genetics[59]. In Canada, similar networks already exist, such as those between the British Columbia Cancer Agency, Family Practice Oncology Network, and Ovarian Cancer Canada. These networks promote continuous professional development, accreditation, and research, while tailoring resources to province-specific needs, ensuring healthcare professionals remain updated on cancer care through educational initiatives[60].

---

## Box 1 | Summary of key recommendations for implementing RSBS

**Defining care pathways**
- If the screening start age is fixed (e.g., 40), use mammographic density from the first screening mammogram, together with PRS and questionnaire-based risk factors, to define the inter-screening interval. If risk assessment occurs earlier and includes only PRS and questionnaire factors, use the resulting risk level to determine the screening start age; then incorporate mammographic density at the first mammogram to refine screening frequency and imaging modalities.
- Set national RSBS strategy with standardized, evidence-informed risk thresholds and screening guidelines. This includes standardized yet flexible care pathways that define roles, responsibilities, and workflows across healthcare professionals, ensuring broad alignment with federal guidelines while being adaptable to regional infrastructure and capacity needs.
- Design care pathways that distribute responsibilities across multidisciplinary teams to reduce the burden on healthcare professionals and promote sustainable delivery.
- Integrate interoperable IT systems into care pathways to enable secure, efficient collection, sharing, and linkage of all risk assessment components and screening recommendations using unique patient identifiers. Align with national interoperability goals and expand infrastructure and resources to meet RSBS needs, including accredited labs, MRI access, workforce training, and public education.
- Ensure targeted strategies help all populations, including visible minority groups, complete risk assessments and access screening equitably. Embed privacy protections and data governance to ensure care pathways are equitable, culturally appropriate, and inclusive of underserved groups.

**Implementation roadmap**
- Establish a national, multidisciplinary implementation group to collaboratively design and oversee the implementation roadmap, ensuring aligned priorities and inclusive stakeholder representation.
- Invest in infrastructure, including accredited genetic laboratories and interoperable IT systems to support risk assessment processes and data linkages.
- Secure sustainable funding for RSBS by aligning it with National Cancer Control Strategies, building a business case, and exploring diverse funding streams, including government, research grants, and private partnerships.

- Adopt a phased, stepwise implementation approach, starting with pilot studies to assess feasibility, and guide tailored rollout across jurisdictions.
- Embed real-time monitoring and evaluation and adopt a Learning Health System (LHS) approach to support continuous, iterative improvement during implementation.

**Stakeholder engagement**
- Establish a national coordination body (e.g., led by Canadian Partnership Against Cancer (CPAC) or through Canadian Breast Cancer Screening Networks (CBCSN)) to oversee stakeholder engagement efforts and ensure continuity, trust, shared ownership, and collaboration across jurisdictions and levels.
- Conduct stakeholder mapping early in the planning stage to identify all relevant stakeholder groups at all levels (e.g., government, healthcare, community, funders, policy, and the public) and refine the map as roles and priorities evolve during implementation. Ensure inclusive engagement of ethnic minorities and marginalized communities.
- Develop engagement objectives and tailor approaches based on stakeholder characteristics, such as influence, interest, capacity, and expertise, using co-design and consultation methods where appropriate.
- Continuously evaluate engagement efforts to stay responsive to changing stakeholder perspectives, emerging evidence, and evolving implementation needs.

**Education and training for healthcare professionals**
- Leverage and adapt existing training platforms (e.g., CME's and MOOC's) to co-design flexible and accessible RSBS-specific resources (e.g., e-learning modules, mobile apps) on risk assessment, genomics, and risk communication tailored to healthcare professionals' time and capacity.
- Conduct workforce mapping and training needs assessments to identify key healthcare provider groups, their knowledge gaps, and specialty-specific needs.
- Build momentum by raising early awareness through foundational education, while developing specialized training in parallel as evidence and care pathways are finalized.

Abbreviations: RSBS risk-stratified breast screening, PRS polygenic risk score, P/T province/territory, MRI magnetic resonance, IT information technology

---

Potential resistance to change among healthcare professionals was identified as one potential risk. Participants noted that education and training strategies would occur later in the roadmap, following early planning stages like defining care pathways. To mitigate this, they recommended a parallel approach that raises broad awareness of RSBS early to build momentum across medical communities. They also suggested involving healthcare professionals in designing educational resources to ensure they reflect their time constraints, capacity, and role in the program.

### Action plan for education and training healthcare professionals

Training healthcare professionals in RSBS would begin with a documentation analysis to map the clinical workforce landscape. An education and training needs assessment would then identify existing knowledge gaps across professional specialties[54] guiding the content and format of educational materials.

Training could be designed and structured at two levels. The first would focus on healthcare professionals already working in clinical roles, such as PCPs and radiologists, providing specialized training for interpreting and communicating risk results[36]. The second would involve integrating genomics education into medical curricula, ensuring future professionals graduate with a broader understanding of genomics[54,61], and are prepared to incorporate RSBS into routine care. Two training strategies could support these levels. Firstly, a planned, course-based approach would provide structured education through online or in-person modules to develop essential RSBS skills. The second is an on-demand, "just-in-time" model, to provide quick, accessible decision-support tools that healthcare practitioners could use during patient appointments to manage or review complex or unfamiliar risk profiles. Importantly, education resources should be regularly updated to reflect evolving scientific evidence, clinical practices, and the changing needs and preferences of healthcare professionals across diverse specialties.

Participants also emphasized the importance of involving a broad range of stakeholders, including healthcare professionals, in developing and updating training tools [see section on Stakeholder Engagement]. This collaborative approach ensures training curricula remain relevant, practical, and effective for users. Similar findings have been reported in other studies involving healthcare professionals[17,59].

## Future directions

This paper focuses on four key areas selected by workshop participants based on their perceived importance and feasibility (See Box 1 for key recommendations). Given the time-limited 2-day workshop format, discussions had to be prioritized, and as a result, some relevant implementation domains could not be explored in detail. Furthermore, while participants represented jurisdictions across Canada, we had one to two representatives from most provinces and territories for logistical reasons. Therefore, the workshop outputs provide a pan-Canadian overview but may not capture the full range of jurisdiction-specific perspectives and do not replace detailed planning within each jurisdiction.

However, actionable plans for implementing RSBS in practice would require attention to several additional considerations. Planning should prioritize health equity to ensure RSBS is accessible and inclusive for all populations[62], including First Nations, Inuit, and Métis peoples, individuals in rural and remote areas, people with varying literacy levels, and those from diverse cultural backgrounds[63]. Intentional strategies will be important to prevent worsening existing health disparities and to promote equitable participation[63,64]. Legal and data regulatory frameworks should be developed to protect individuals from genetic discrimination and stigmatization[20,21], including provisions for supporting those who choose not to participate in RSBS programs[26]. In parallel, communication strategies should be carefully designed to deliver clear, consistent messaging around risk assessment processes and follow-up actions, fostering patient understanding, trust, and continued engagement[25]. Establishing RSBS as a government priority by embedding it into national health strategies (e.g., Cancer Care Control Strategy) would require planning efforts. A supportive political climate is critical to secure government commitment[65], sustained funding, and long-term support for planning, implementation, and evaluation of an RSBS program. Finally, it will be important to remain open and flexible to opportunities for leveraging emerging technologies. Advances such as AI for risk prediction[66] could help with risk prediction and identify optimal screening intervals, which would also reduce workload in reading mammograms, and support in collecting risk information from EMR[67]. Similarly, advances in low-dose mammography and deep-learning techniques for accurate breast density estimation at significantly reduced radiation exposure[68] could also be incorporated to support future RSBS programs.

While implementation should be context-specific, these workshop insights can inform international planning on the logistical and organizational aspects of an RSBS program, particularly in settings with comparable healthcare systems. Our workshop outputs distinguish between the core functions needed to implement RSBS and the local forms those functions may take. In Canada, several recommendations are shaped by the organization of screening across provincial and territorial programs and by the roles of national/provincial bodies in governance, data-sharing, and guideline development; however, the underlying principles and action areas reflected in our key recommendations (Box 1) are broadly applicable across jurisdictions. These include defining clear care pathways, determining which risk information to collect, establishing thresholds for risk stratification, early and sustained multi-stakeholder engagement, equity-by-design, interoperable data and evaluation planning, and proactive education and training of the clinical workforce. While the specific pathways, thresholds, risk categories, risk factors included in assessment, and choice of risk assessment tools will vary based on local screening infrastructure, governance models, and population needs, the core steps are transferable and can be adapted to support implementation planning in other settings.

## Conclusion

There is no universal gold standard for implementing an RSBS program. What is important, however, is an early and deliberate planning strategy that identifies context-specific needs and challenges while leveraging existing strengths and opportunities. Such tailored planning not only supports effective local implementation but also generates insights that can be shared across jurisdictions to promote broader learning, improve population health outcomes, and enhance system efficiency. Implementation should not be viewed as a linear process with a fixed endpoint. Rather, it requires ongoing learning, adaptation, and integration of new evidence to continually refine workflows and strategies. A learning healthcare system approach supports this adaptive planning by viewing the health system as a dynamic, interconnected whole. This approach emphasizes cross-stakeholder level collaboration and encourages learning by doing through ongoing monitoring, feedback, and adaptation to guide decision-making and improve implementation over time.

## Data availability

Workshop session recordings and transcripts are not publicly available because they contain information that could risk re-identification of participants. A de-identified synthesis of workshop outputs (e.g., summary posters/aggregated materials) is available from the corresponding author upon reasonable request.

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

## Acknowledgements

We would like to acknowledge the *Quebec Breast Cancer Foundation* for their financial support for the workshop organization. We would also like to acknowledge the *Canadian Partnership Against Cancer* for their support in facilitating the participation of the Canadian Breast Cancer Screening Network in the workshop. The Canadian Partnership Against Cancer's contribution was made possible through financial support from Health Canada. We also thank Mélissa Côté (*Institut national d'excellence en santé et en services sociaux, Quebec City, QC, Canada*), Marie-Hélène Guertin (*Bureau d'information et d'études en santé des populations, Institut national de santé publique du Québec, Quebec City, QC, Canada*), Nicole Mitmann (*Sunnybrook Research Institute, Sunnybrook Health Sciences Centre, Toronto, ON, Canada*), and Trena Metcalfe (*Nova Scotia Breast Screening Program, IWK Health, Halifax, NS, Canada)* for their contributions. This workshop formed part of the PERSPECTIVE I&I project, funded by the Quebec Breast Cancer Foundation and supported by the Government of Canada through Genome Canada (grant no: 13529), the Canadian Institutes of Health Research (grant no: 155865), the Ministère de l'Économie et de l'Innovation du Québec through Génome Québec, the Quebec Breast Cancer Foundation, the CHU de Québec Foundation, and the Ontario Ministry of Research and Innovation through the Ontario Research Fund. A.C.A. is supported by Cancer Research grants UK PPRPGM-Nov20\100002 and SEBCD3-2024/100001.

## Author contributions

N.S.G.: Conceptualization, Methodology/Workshop design, Investigation—Workshop Facilitation, Synthesis, Writing—original draft, Writing—reviewing and editing. A.C.A.: Investigation—Workshop facilitation, Writing—reviewing and editing. J.B.: Investigation— Workshop facilitation, Writing—reviewing and editing. J.Ca.: Workshop participation, Writing—reviewing and editing. M.C.: Workshop participation, Writing—Reviewing and editing. J.Ch.: Workshop participation, Writing—reviewing and editing. M.D.: Workshop participation, Writing—Reviewing and editing. G.D.: Workshop participation, Writing—Reviewing and editing. A.E.: Workshop participation, Writing—Reviewing and editing. Laurence Eloy: Workshop participation, Writing—Reviewing and editing. S.F.: Workshop participation, Writing—Reviewing and editing. V.F.: Workshop participation, Writing—Reviewing and editing. K.H.: Workshop participation, Writing—Reviewing and editing. S.K.: Workshop participation, Writing—Reviewing and editing. C.Mb.: Workshop participation, Writing—Reviewing and editing. C.Mo.: Workshop participation, Writing—Reviewing and editing. H.N.: Investigation—Workshop facilitation, Writing—Reviewing and editing. C.S.: Workshop participation, Writing—Reviewing and editing. J.M.S.: Workshop Participation, Writing—Reviewing and editing. A.J.S.: Workshop participation, Writing—Reviewing and editing. P.S.: Methodology/workshop design, Writing—Reviewing and editing. R.T.: Workshop participation, Writing—Reviewing and editing. I.T.: Workshop participation, Writing—Reviewing and editing. A.T.: Methodology/workshop design, Writing—Reviewing and editing. M.V.: Workshop participation, Writing—Reviewing and editing. N.W.: Workshop participation, Writing—Reviewing and editing. M.Wa.: Investigation—Workshop facilitation, Writing—reviewing and editing. M.Wo.: Workshop participation, Writing—Reviewing and editing. A.C.: Methodology/workshop design, Funding acquisition, Investigation—Workshop facilitation, Writing—Reviewing and editing. J.S.: Conceptualization, Funding acquisition, Methodology/workshop design, Investigation—Workshop facilitation, Writing—Reviewing and editing, Supervision. N.P.: Conceptualization, Methodology/workshop design, Investigation—Workshop facilitation, Synthesis, Writing—Original draft, Writing—reviewing and editing, Supervision. All authors read and approved the final version of the manuscript.

## Competing interests

Antonis C. Antoniou is listed as the creator of the BOADICEA model, which has been licensed by Cambridge Enterprise, University of Cambridge. MJ DeCoteau has Advisory board participation honoraria from: Gilead Sciences, Hoffman-La Roche, and Novartis. All other authors declare no competing interests.

## Additional information

[1]Centre for Cancer Genetic Epidemiology, Department of Public Health and Primary Care, University of Cambridge, Cambridge, UK. [2]Dalla Lana School of Public Health, University of Toronto, Toronto, ON, Canada. [3]Ray D. Wolfe Department of Family Medicine, Mount Sinai Hospital, Sinai Health, Toronto, ON, Canada. [4]Department of Family and Community Medicine, University of Toronto, Toronto, ON, Canada. [5]Department of Pediatrics, Faculty of Medicine, Université Laval, Quebec City, QC, Canada. [6]Oncogenetic Services, CHU de Québec–Université Laval, Quebec City, QC, Canada. [7]CHU de Québec–Université Laval Research Center, Quebec City, QC, Canada. [8]Département de médecine de famille et de médecine d'urgence, Université Laval, Quebec City, QC, Canada. [9]Rethink Breast Cancer, Toronto, ON, Canada. [10]Cancer Care, Newfoundland and Labrador Health Services, St. John's, NL, Canada. [11]Ontario Health, Toronto, ON, Canada. [12]Sunnybrook Research Institute, Sunnybrook Health Sciences Centre, Toronto, ON, Canada. [13]Programme québécois de cancérologie, Ministère de la Santé et des Services sociaux, Quebec City, QC, Canada. [14]Centre intégré de santé et de services sociaux de Lanaudière, Joliette, QC, Canada. [15]Joint Department of Medical Imaging, University of Toronto; University Health Network; Sinai Health System; Women's College Hospital, Toronto, ON, Canada. [16]Diagnostic Imaging (Radiology), Queen Elizabeth Hospital (QEH), Health PEI, Charlottetown, PEI, Canada. [17]Department of Social and Preventive Medicine, Faculty of Medicine, Université Laval, Quebec City, QC, Canada. [18]Department of Radiology, Cumming School of Medicine, University of Calgary, Calgary, AB, Canada. [19]Department of Radiology, Radio-Oncology and Nuclear Medicine, Faculty of Medicine, University of Montreal, Montreal, QC, Canada. [20]Department of Radiology, University of Ottawa, Ottawa, ON, Canada. [21]Indigenous Health Unit, Ontario Health, Toronto, ON, Canada. [22]Genomics Center, CHU de Québec–Université Laval Research Center, Quebec City, QC, Canada. [23]Department of Health, Government of Nunavut, Iqaluit, NU, Canada. [24]Department of Radiology, Centre hospitalier de l'Université de Montréal (CHUM), Montréal, QC, Canada. [25]Canadian Association of Radiologists, Ottawa, ON, Canada. [26]Faculty of Medicine, Memorial University of Newfoundland, St. John's, NL, Canada. [27]University of Ottawa Faculty of Medicine, Ontario, ON, Canada. [28]Department of Molecular Medicine, Faculty of Medicine, Université Laval, Quebec City, Canada. [29]These authors contributed equally: Jacques Simard, Nora Pashayan. ✉e-mail: ns951@cam.ac.uk; jacques.simard@crchudequebec.ulaval.ca; np275@medschl.ca.ac.uk

