## [Transparent Peer Review File · Communications Medicine]

Perspectives from a multi-stakeholder workshop for implementing a risk-stratified breast cancer screening program in Canada

Corresponding Author: Ms Nareh Safieh-Garabedian

Version 0:

Reviewer comments:

Reviewer #1

(Remarks to the Author)

This paper reports findings from a multi-stakeholder workshop on the planning of implementing population-wide risk-stratified breast screening. The topic of risk-stratified screening is timely and relevant internationally, as the next steps for breast cancer screening in order to maximise benefits and reduce harms. Screening programs around the world are interested and looking towards making such changes, however implementation is not straight forward. This paper provides a comprehensive summary of the workshop outputs for next steps in regard to implementation in Canada.

While the paper is well written and these steps are important to understand, my main comment is that there needs to be greater recognition of the distinctness of the key recommendations for implementation for Canada and that these may not be relevant for other jurisdictions as all screening programs and healthcare setting internationally vary. Otherwise, the piece lacks generalisability for those outside of Canada. Suggest including more discussion around this where possible throughout including in the future directions section.

Other comments:

- Suggest having supplementary Figure 1 in a table instead of a figure with other demographics as well. This table could also be part of the main paper and not supplement although I understand this is not a traditional research paper.
- The tables are a very helpful summary of the paper's text which feels quite long and in places quite heavy. I wonder whether some of the text could be condense a bit to reduce repetition or there could be numbers or something to make the main body of the text a bit easier to follow/break it up.
- The figures need to be bigger – currently quite hard to read.
- There are no limitations to the method discussed.

Reviewer #2

(Remarks to the Author)

1. Brief summary of the manuscript

Thank you for asking me to review this most interesting manuscript which reports the findings of a workshop that was convened with 35 participants who were tasked with envisioning how a risk stratified breast screening programme could be integrated into the existing healthcare system. The manuscript is structured around the four key topics identified and attempts to produce a useful actionable plan/steps for each topic. The paper concludes with a set of key recommendations in each topic area selected and recognises that there are topics that were not considered during this workshop identifying a number of these in the 'future directions section'.

2. Overall impression of the work

Overall I found the manuscript well-written and clearly structured. Where possible the authors have attempted to be constructive and put forward practical tables (plans) and checklists which can be utilised by other parties who interested in these topic areas. I was impressed with the paper overall and consider that there are very few revisions required. To this end most of my comments are of optional interest rather than prescriptive.

3. Specific comments, with recommendations for addressing each comment

Introduction:

- a) This section sets the background and context for the possible introduction of RSBS. The context is Canada and as a person from outside the area I would be interested to understand a) the size of the programme (invitations per year nationally) and perhaps uptake/response levels
- b) Line 164 refers to the justifications for RSBS (e.g. clear justification in terms of balance of benefit/harm). Is there any indication that RSBS will be mandated in Canada any time soon?

Workshop design:

The design is well described and looks to be relevant and geographically diverse. You enumerate the participants in each category.

- a) Given that health equity is of importance are you able to indicate the ethnic mix and gender mix of participants? I did note that there was just one patient/public participant.
- b) Do you have any suggested improvements for the workshop design itself? If we were attempting to carry out a similar exercise, could we learn from any difficulties you had? If so, any suggestions probably belong in the final section of the manuscript.

Mapping the pathway for a RSBS program

- a) The Care pathway flow charts Figure 1 are quite difficult to read. Hopefully when published these can be provided at a larger scale and font.
- b) Was consideration given to how/whether to integrate those already identified as high risk gene carriers and already on a more intensive/tailored care pathway? What would happen to those who decline the risk assessment? It may be worth noting that only the key pathways are illustrated
- c) Do you envisage national guidelines that will specify recommended screening frequency/modality for different age groups and risk levels?
- d) Line 329, 392, 460, 527 – It would be helpful if the section header 'Action Plan' could be expanded to say Action plan for topic. Would aid document navigation for the reader
- Table 1 was a nice way to compare the existing systems with the future needs and to attempt a classification of the challenges to be faced.

Implementation Roadmap:

I found this section very helpful and the action plan table nicely structured. It provides a useful starting point for similar work in a different country context.

Education and training for healthcare professionals:

Similarly, this was a useful section highlighting the need to be ahead of the curve e.g. by raising broad awareness before the finalised pathways are available. Your action plan table forms a useful starting point.

4. What are the major claims of the paper?

The paper aims to produce practical output that serves as a starting point that is also transferable to other RSBS contexts. I believe that it meets these aims.

5. Will the paper be of interest to others in the field?

The paper will be of interest and is highly pertinent given increased interest in RSBS

6. Will the paper influence thinking in the field?

I believe that the paper will influence and support similar activities in different screening programmes and in different countries

Version 1:

Reviewer comments:

Reviewer #1

(Remarks to the Author)

The authors have adequately addressed all of my and the other reviewers' previous comments and concerns and the manuscript has been edited accordingly. I have nothing further to add.

Reviewer #2

(Remarks to the Author)

I am happy that all my concerns have been addressed

Authors' response to the Reviewers' comments

Manuscript COMMSMED-25-2784: Perspectives from a multi-stakeholder workshop for implementing a risk-stratified breast cancer screening program in Canada

We appreciate the opportunity to submit a revised version of our manuscript (COMMSMED-25-2784), "Perspectives from a multi-stakeholder workshop for implementing a risk-stratified breast cancer screening program in Canada". We thank the reviewers for their constructive comments and provide our point-by-point responses below (in blue font).

Reviewers' comments:

Reviewer #1

This paper reports findings from a multi-stakeholder workshop on the planning of implementing population-wide risk-stratified breast screening. The topic of risk-stratified screening is timely and relevant internationally, as the next steps for breast cancer screening in order to maximise benefits and reduce harms. Screening programs around the world are interested and looking towards making such changes, however implementation is not straight forward. This paper provides a comprehensive summary of the workshop outputs for next steps in regard to implementation in Canada.

While the paper is well written and these steps are important to understand, my main comment is that there needs to be greater recognition of the distinctness of the key recommendations for implementation for Canada and that these may not be relevant for other jurisdictions as all screening programs and healthcare setting internationally vary. Otherwise, the piece lacks generalisability for those outside of Canada. Suggest including more discussion around this where possible throughout including in the future directions section.

Thank you for this feedback. We agree and have revised the manuscript to more clearly distinguish Canada-specific implementation considerations from transferable insights. We added the following text to the "Future directions" section (p. 19; paragraph 2):

"Our workshop outputs distinguish between the core functions needed to implement RSBS and the local forms those functions may take. In Canada, several recommendations are shaped by the organization of screening across provincial and territorial programs and by the roles of national/provincial bodies in governance, data-sharing, and guideline development, however, the underlying principles and action areas reflected in our key recommendations (**Table 4**) are broadly applicable across jurisdictions. These include defining clear care pathways, determining which risk information to collect, establishing thresholds for risk stratification, early and sustained multi-stakeholder engagement, equity-by-design, interoperable data and evaluation planning, and proactive education and training of the clinical workforce. While the specific pathways, thresholds, risk categories, risk factors included in assessment, and choice of risk assessment tools will vary based on local screening infrastructure, governance models, and population needs, the core steps are transferable and can be adapted to support implementation planning in other settings."

Other comments:

- Suggest having supplementary Figure 1 in a table instead of a figure with other demographics as well. This table could also be part of the main paper and not supplement although I understand this is not a traditional research paper.

Thank you for this suggestion. We removed the supplementary figure and replaced it with a participant demographics table (**Supplementary Table 2**), which provides a more complete summary of workshop attendee characteristics.

- The tables are a very helpful summary of the paper's text which feels quite long and in places quite heavy. I wonder whether some of the text could be condense a bit to reduce repetition or there could be numbers or something to make the main body of the text a bit easier to follow/break it up.

Thank you for this feedback. We have revised the tables to reduce text heaviness where possible. We adjusted the layout of **Table 1** to a landscape format, reformatted sections of the table, and cut down wording to shorten the text. We also divided **Table 1** by breaking it down into three components so it can be navigated more easily. The biggest changes were to **Table 1**, although we also cut few words where possible in the other tables to reduce text heaviness.

- The figures need to be bigger – currently quite hard to read.

Thank you for this suggestion. We increased the size and font of all figures.

- There are no limitations to the method discussed.

Thank you for this feedback. We added a limitation to the section "Future directions" (p.18; paragraph 4):

"Given the time-limited 2-day workshop format, discussions had to be prioritized, and as a result, some relevant implementation domains could not be explored in detail. Furthermore, while participants represented jurisdictions across Canada, we had one to two representatives from most provinces and territories for logistical reasons. Therefore, the workshop outputs provide a pan-Canadian overview but may not capture the full range of jurisdiction-specific perspectives and do not replace detailed planning within each jurisdiction."

Reviewer #2 (Remarks to the Author):

1. Brief summary of the manuscript

Thank you for asking me to review this most interesting manuscript which reports the findings of a workshop that was convened with 35 participants who were tasked with envisioning how a risk stratified breast screening programme could be integrated into the existing healthcare system. The manuscript is structured around the four key topics identified and attempts to produce a useful actionable plan/steps for each topic. The paper concludes with a set of key recommendations in each topic area selected and recognises that there are topics that were not considered during this workshop identifying a number of these in the 'future directions section'.

Thank you for reviewing our manuscript.

2. Overall impression of the work

Overall I found the manuscript well-written and clearly structured. Where possible the authors have attempted to be constructive and put forward practical tables (plans) and checklists which can be utilised by other parties who interested in these topic areas. I was impressed with the paper overall and consider that there are very few revisions required. To this end most of my comments are of optional interest rather than prescriptive.

Thank you very much for this feedback.

3. Specific comments, with recommendations for addressing each comment

Introduction:

a) This section sets the background and context for the possible introduction of RSBS. The context is Canada and as a person from outside the area I would interested to understand a) the size of the programme (invitations per year nationally) and perhaps uptake/response levels

Thank you for this suggestion. We did not include a single national uptake estimate because participation metrics vary and are reported differently across jurisdictions and time periods. We added brief context on the organization of population-based screening across provincial and territorial programs with the following text added to the "Introduction" section (p. 5; paragraph 1):

"Guided by national recommendations, organized breast screening is delivered through provincial/territorial programs, each with its own policies and operational models (e.g., eligibility, entry routes, data systems, and follow-up pathways). All provinces and territories have implemented population-wide breast screening programs, except Nunavut, where breast screening is offered opportunistically, and some jurisdictions have established high-risk pathways (e.g., the High-Risk Ontario Breast Screening Program)."

b) Line 164 refers to the justifications for RSBS (e.g. clear justification in terms balance of benefit/harm). Is there any indication that RSBS will be mandated in Canada any time soon?

Thank you for this question. As breast screening is delivered through provincial and territorial programs, any move toward adopting RSBS would likely be jurisdiction-specific and evidence-informed. There are indications of policy interest in risk-based approaches in some jurisdictions - for example, the National Assembly of Québec have unanimously adopted a motion supporting integration of a risk-based screening approach.

[Assemblée nationale du Québec. *Journal des débats* (43e législature, 2e session). Motion supporting integration of a risk-based screening approach. 2 Oct 2025. Available at:

<https://www.assnat.qc.ca/fr/travaux-parlementaires/assemblee-nationale/43-2/journal-debats/20251002/416705.html>]

Workshop design:

The design is well described and looks to be relevant and geographically diverse. You enumerate the participants in each category.

a) Given that health equity is of importance are you able to indicate the ethnic mix and gender mix of participants? I did note that there as just one patient/public participant.

Thank you for this feedback. We acknowledge the importance of reporting participant characteristics through a health equity lens. Self-identified ethnicity information was not collected and therefore cannot be reported. We have added a demographics table (**Supplementary Table 2**) summarizing participants' professional role, province/territory, and gender. While one participant attended as a patient/public participant, two participants also represented patient/public organizations; we have clarified this in the "Workshop Design" section (p. 6; paragraph 4):

"Many participants held multiple roles, including individuals working in provincial governments (n=14), managers or directors of jurisdictional breast screening programs (n=17), and healthcare practitioners (n=8), including family physicians and radiologists. One participant attended as a patient/public advocate, and two participants also represented patient/public organizations, including the Quebec Breast Cancer Foundation and Rethink Breast Cancer."

b) Do you have any suggested improvements for the workshop design itself? If we were attempting to carry out a similar exercise, could we learn from any difficulties you had? If so this any suggestions probably belong in the final section of the manuscript.

The workshop format enabled rapid, multi-stakeholder input; however, a limitation was the time constraints of 2 days which limited the possibility of exploring additional implementation domains. Please refer to our response above to reviewer 1 where we address this by adding some limitations to the section "Future directions" (p.18; paragraph 4).

Mapping the pathway for a RSBS program

a) The Care pathway flow charts Figure 1 are quite difficult to read. Hopefully when published these can be provided at a larger scale and font.

Thank you for this feedback. We revised **Figures 1 and 2** to improve readability by increasing the overall size and font.

b) Was consideration given to how/whether to integrate those already identified as high risk gene carriers and already on a more intensive/tailored care pathway?

We agree this is an important consideration. We clarified that existing high-risk identification and management pathways (e.g., known *BRCA1/BRCA2* pathogenic variant carriers) are distinct from the population-wide RSBS pathways described here, that high-risk services vary across provinces/territories, and that **Figure 1** illustrates key RSBS pathways only (high-risk pathways were outside the scope of the workshop). We added the following text to "Mapping the pathway for a risk-stratified breast screening (RSBS) program" (p. 9, paragraph 2):

"Participants also noted that existing high-risk identification and management pathways (e.g., for individuals with known pathogenic variants such as *BRCA1/BRCA2*) are distinct from the population-wide RSBS pathways described here. As high-risk identification and services differ across provinces and territories in Canada (formal programs vs pathways), **Figure 1** illustrates key pathways for RSBS and is not intended to be exhaustive. High-risk pathways were not mapped in detail as this was

outside the scope of the workshop discussions, although participants considered this as an important topic area for future work.”

What would happen to those who decline the risk assessment? It may be worth noting that only the key pathways are illustrated

Thank you for raising this important point, which was also discussed by workshop participants. We added the following text to the “Mapping the pathway for a risk-stratified breast screening (RSBS) program” section (p. 9; paragraph 2):

“Workshop participants emphasized that individuals should have the option to decline risk assessment. In such cases, an alternative pathway should be available such as sustained access to standard age-based screening (with appropriate information and the option to reconsider risk assessment in the future)”.

c) Do you envisage national guidelines that will specify recommended screening frequency/modality for different age groups and risk levels?

Yes. We envisage that evidence-informed guidance will be needed to specify screening frequency and modality by age group and risk level. In Canada, this may involve pan-Canadian alignment on key elements (e.g., risk thresholds and recommended modalities/intervals), while operational implementation would remain the responsibility of provincial and territorial programs. The timing and content of such guidance would depend on the emerging evidence base and on jurisdictional policy processes.

d) Line 329, 392, 460,527 – It would be helpful if the section header ‘Action Plan’ could be expanded to say Action plan for topic. Would aid document navigation for the reader

Thank you for this suggestion. We revised the headers to specify the topic area for each action plan.

Table 1 was a nice way to compare the existing systems with the future needs and to attempt a classification of the challenges to be faced.

Thank you for this feedback.

Implementation Roadmap:

I found this section very helpful and the action plan table nicely structured. It provides a useful starting point for similar work in a different country context.

Thank you for this feedback.

Education and training for healthcare professionals:

Similarly, this was a useful section highlighting the need to be ahead of the curve e.g. by raising broad awareness before the finalised pathways are available. Your action plan table forms a useful starting point.

Thank you for this feedback.

4. What are the major claims of the paper?

The paper aims to produce practical output that serves as starting point that is also transferable to other RSBS contexts. I believe that it meets these aims.

5. Will the paper be of interest to others in the field?

The paper will be of interest and is highly pertinent given increased interest in RSBS

6. Will the paper influence thinking in the field?

I believe that the paper will influence and support similar activities in different screening programmes and in different countries

We thank the editor and reviewers very much for their time and consideration of our manuscript for publication in *Communications Medicine*.

Yours sincerely,

Prof Nora Pashayan, MD, PhD, FFPH

Prof Jacques Simard, PhD

Nareh Safieh-Garabedian BMus, MA